# Niche-Based Microbial Community Assemblage in Urban Transit Systems and the Influence of City Characteristics

Guangzhou Xiong,[a] Lei Ji,[a] Mingyue Cheng,[a] Kang Ning[a]

[a]Key Laboratory of Molecular Biophysics of the Ministry of Education, Hubei Key Laboratory of Bioinformatics and Molecular-imaging, Center of AI Biology, Department of Bioinformatics and Systems Biology, College of Life Science and Technology, Huazhong University of Science and Technology, Wuhan, Hubei, China

**ABSTRACT**   Microbiota residing on the urban transit systems (UTSs) can be shared by travelers and have niche-specific assemblage. However, it remains unclear how the assemblages are influenced by city characteristics, rendering city-specific and microbial-aware urban planning challenging. Here, we analyzed 3,359 UTS microbial samples collected from 16 cities around the world. We found the stochastic process dominated in all UTS microbiota assemblages, with the explanation rate ($R^2$) of the neutral community model (NCM) higher than 0.7. Moreover, city characteristics predominantly drove such assemblage, largely responsible for the variation in the stochasticity ratio (50.1%). Furthermore, by utilizing an artificial intelligence model, we quantified the ability of UTS microbes in discriminating between cities and found that the ability was also strongly affected by city characteristics, especially climate and continent. From these, we found that although the NCM $R^2$ of the New York City UTS microbiota was 0.831, the accuracy of the microbial-based city characteristic classifier was higher than 0.9. This is the first study to demonstrate the effects of city characteristics on the UTS microbiota assemblage, paving the way for city-specific and microbial-aware applications.

**IMPORTANCE**   We analyzed the urban transit system microbiota assemblage across 16 cities. The stochastic process was dominant in the urban transit system microbiota assemblage. The urban transit system microbe's ability in discriminating between cities was quantified using transfer learning based on random forest (RF) methods. Certain urban transit system microbes were strongly affected by city characteristics.

**KEYWORDS**   urban transit system, artificial intelligence, microbiota assemblage, city characteristics

Among the hundreds of biomes, 90% of human interactions are within built environments (BEs) (1) due to urbanization in the last 150 years. Thus, humans are mostly exposed to the complex BE microbial communities in daily life and exchange their microbiota (2, 3) within BEs through common activities such as skin contact (4), respiratory activity (5, 6), and skin shedding (7). Microbes associated with increasing urbanization (8) in the built environment have been implicated as a possible source of contagion (9) and certain syndromes, such as allergies (10). Studies of the urban built environment microbiomes have spanned several different projects and initiatives, including work focused on transit systems (11–15), hospitals (16, 17), soil (18, 19), and sewage (20, 21). However, only a few selected cities were studied on a limited number of occasions for the most part. This leaves a gap in scientific knowledge about a microbial ecosystem with which the global human population readily interacts. Human commensal microbiomes have also been found to vary based on culture, and thus, geographically isolated studies are limited and may miss key differences (22). Indeed, our understanding of microbial dynamics in the urban environment outside pandemics has just begun (23).

Urban transit systems (UTSs) represent daily contact interfaces for millions of people who live in cities. Urban travelers share their commensal microorganisms with other

Address correspondence to Kang Ning, ningkang@hust.edu.cn.

The authors declare no conflict of interest.

passengers as they travel and come into contact with organisms and mobile elements present in the environment (24). Thus, mass transit environments such as subways facilitate a constant flow of microbes among humans and different BEs (12). UTSs are thus particularly important for human health due to their potential for spreading pathogens (25) and impacting large numbers of people. For example, coronavirus disease 2019 (COVID-19)-asymptomatic subjects could mass transmit the pathogenic severe acute respiratory syndrome coronavirus (SARS-CoV-2) in UTSs by breathing (26). Although the COVID-19 pandemic had an immediate and significant impact on public transportation ridership, ridership declined to a low point of 10% to 40% of prepandemic levels in many cities in the United States (27). The United States had 4.7 billion unlinked transit passenger trips in 2021 (20 January to 20 December) (28). The diversity, composition, and stability of UTS microbiota are influenced by multiple factors (7), including surface type (12) and city characteristics (including latitude, longitude, altitude, and climate). Therefore, understanding the assemblage of UTS microbiota on a global scale, as well as mining the association between UTS microbiota and city characteristics, is of great importance for deep understanding of UTS microbiota diversity and evolution and can inform decisions regarding public health.

In this work, we aim to gain a better understanding of UTS microbiota, specifically to explore whether the UTS microbiota assemblage was dominated by deterministic or stochastic processes, whether the assemblage was influenced by city characteristics, and to what extent the city characteristics could shape city-specific microbial community profiles. To answer these questions, the UTS samples ($n = 3,359$) from 16 cities around the world collected by the Metagenomics and Metadesign of the Subways and Urban Biomes (MetaSUB) International Consortium (24) were used in this study. First, the assemblage in shaping the UTS microbiota was examined, and the city characteristics that influence UTS microbiota assemblage were explored and quantified. Second, the ability of UTS microbes in discriminating between cities was quantified by an artificial intelligence model based on transfer learning and random forest (RF) methods. Third, the association between city characteristics and the UTS microbes was established. The results show that the stochastic process is dominant in UTS microbiota assemblage, while the stochastic process was strongly influenced by city characteristics. Also, city characteristics could shape certain city-specific UTS microbes, and, in turn, city characteristics could be accurately predicted by using the microbes. Similar to the precision medicine paradigm, the patterns that we discovered could facilitate a broad spectrum of applications, including public health monitoring, as well as city-specific and microbial-aware urban planning.

## RESULTS

**Assemblage of UTS microbiota.** To investigate the stochastic processes in the urban transit system (UTS) microbiota assemblage, we employed the neutral community model (NCM) to estimate the relationship between the occurrence frequency of genera and the change in relative abundance (Fig. 1). Each city's explanation rates ($R^2$), which indicated the fit of the NCM for the UTS microbiota, exceeded 0.75, except Barcelona (0.721), Fairbanks (0.74), and Santiago (0.714), indicating that the stochastic processes played a dominant role in community assemblage (Fig. 1). Neutral theories predict that random patterns in microbial co-occurrence and environmentally independent spatial autocorrelation (for example, dispersal) should be the main features of community structure if demographic stochasticity and limited dispersal alone drive population dynamics (29–32). A constant flow of microbes among humans and different surfaces exist in UTSs. Thus, the stochastic processes shaped the microbial communities in UTSs and were likely due to frequent microbes exchange by human interaction within the UTS environment.

To explore the contact and potential dispersal route of microbes on the UTS surface, we performed source tracking analyses using FEAST. The samples used in source tracking analyses were collected from New York City UTS surface types ($n = 402$, including 14 different surface types) (see Table S2 in the supplemental material). Microbial communities from 1 surface type were used as the sinks, and the samples from 13

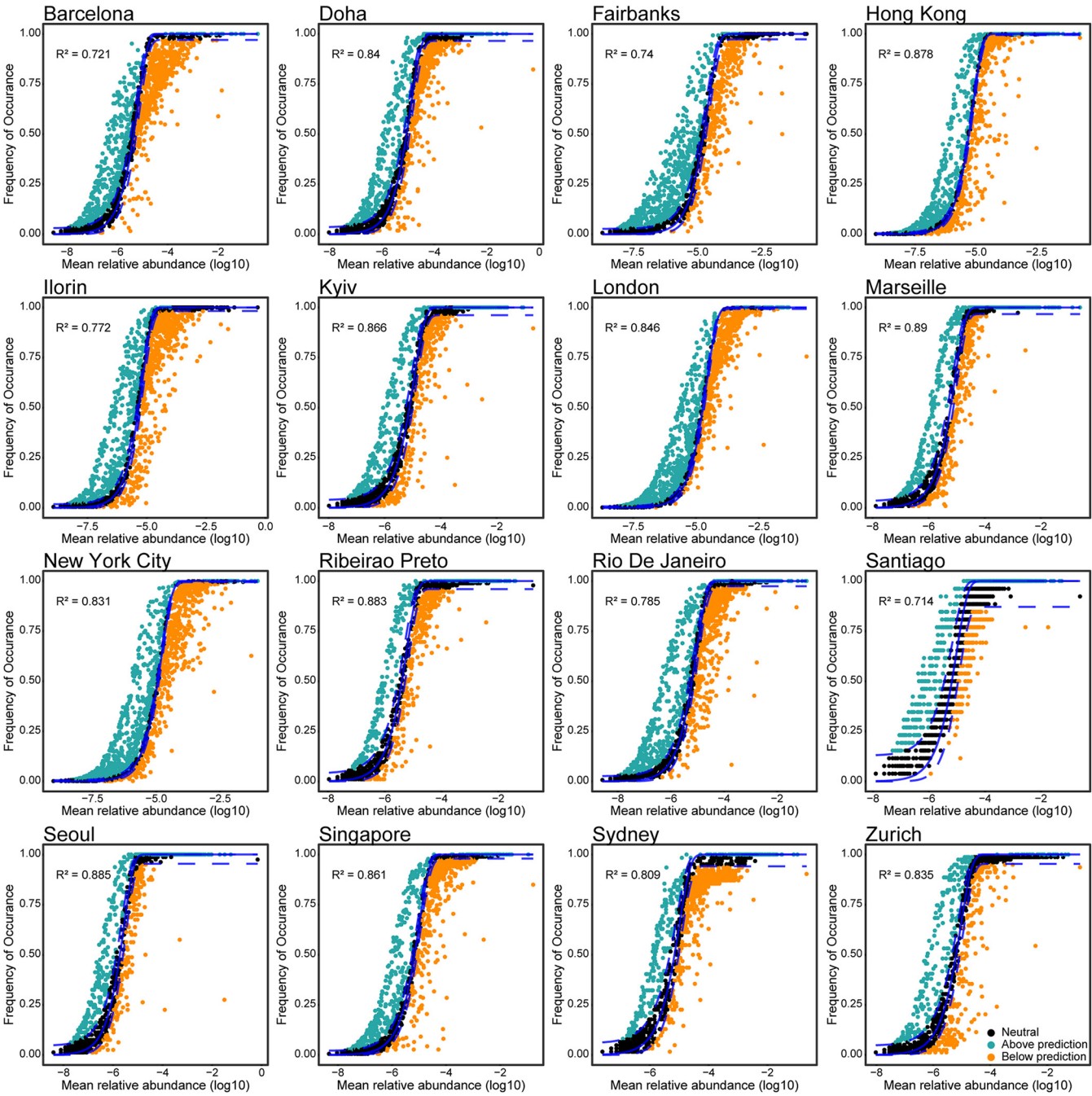

**FIG 1** Assemblage pattern of urban transit system microbial communities. The plots show the fit of the neutral community model (NCM) of urban transit system surface microbial community assemblage of 16 cities around the world. The solid blue line represents the best fit, and the dotted lines represent the 95% confidence interval (CI) around the model prediction. Genera that occur more or less frequently than predicted by the neutral community model are shown in different colors: the blue dots represent the genera above prediction, and the orange dots represent the genera blow prediction. Explanation rate ($R^2$) indicates the fit to NCM.

other surface types were used as the sources. The result of microbial source tracking showed that the kiosk (18.5%) and stairwell railing (14.9%) surface microbial communities were the dominant contributing sources of bench surface microbial communities, and benches (18%) were the dominant contributing source of the kiosk surface microbial communities (Fig. S2A). The unknown source was a small contributor to the four main surface types, which shows that in the UTS, different surface microbial communities may contact internally and spread frequently (Fig. S2B). The result of the source tracking analysis indicated that different surface microbiota are susceptible to dispersal

by frequent microbe exchange, which causes the stochastic process dominant in UTS microbiota assemblage. However, the NCM $R^2$ varied greatly across cities (Fig. 1).

We next sought to explore whether and which city characteristics determined the stochastic process weight in UTS microbiota assemblage. To this end, we calculated the modified stochasticity ratio (MST) (33) to quantify the stochasticity of the assemblage and conducted permutational multivariate analysis of variance (PERMANOVA) to quantify the degree to which city characteristics influenced the MST. We found that city climate was the most important factor (14.7%), followed by the world's overall region (10.9%). The other factors accounted for 24.5% of the variation in MST (Fig. 2A). These results suggested that the city characteristics were strongly associated with the stochastic process weight in UTS microbiota assemblage. Notably, the proximity to the coast and the longitude of the sampled city had no significant effect on MST (Fig. 2A).

The city characteristics may form an ecological filter for certain microbes in UTSs, although the effect varies across cities, and the ecological filter could determine the stochasticity ratio of UTS microbiota (34, 35). To further confirm the ecological filter built by city characteristics in UTSs, we calculated the $\beta_{RC}$ values, which indicated the potential mechanism of community construction of each sample (36). The distribution of $\beta_{RC}$ values in the different UTSs was fairly similar (Fig. 2B); we observed a bimodal distribution of $\beta_{RC}$ values across our data set, and the absolute $\beta_{RC}$ values of the transit system community were close to 1, indicating that the ecological filter was conducive to the composition of different microbes.

The stochastic process that is dominant in the assemblage of UTS surface microbiota decreased the uniqueness of the UTS microbial distribution pattern. However, it was commonly accepted that different cities have different microbial community patterns, regardless of the specific surfaces or locations. Thus, although the assemblage of UTS microbiota was stochastic, discovery of the city-specific UTS microbes is of high importance. However, current methods fall short in accurate identification of city-specific UTS microbes. Therefore, an intelligent data mining method was warranted in discovering the microbiome patterns behind the UTS surface samples. Here, we have evaluated a general framework based on artificial intelligence modeling for quantifying the ability of UTS microbes in discriminating between cities and in proving the strong association between city characteristics and UTS microbes.

**Quantify the city specificity of UTS microbes using artificial intelligence.** Next, we explored the city-specific UTS microbes under the stochastic process of the assemblage. To this end, we introduced an artificial intelligence model, which was built based on transfer learning and random forest. Random forest was commonly used for the discovery of UTS microbes for specific cities, which, however, may not meet the comparison of UTS microbes on the global scale due to the unbalanced number of samples for each city. Transfer learning can help to manage this context to make a fair comparison between samples with unbalanced sizes.

First, we sought to quantify the city specificity of UTS microbes; thus, non-city characteristic influence should be excluded. The interference of surface type, which plays an important role in the possible variation in taxonomy, has been proven previously (12). Thus, we quantify the ability of UTS microbe city specificity for each main surface type (bench, door, handrail, and kiosk) to remove the deviation introduced by surface type.

Second, to improve the accuracy of the UTS microbe city specificity quantifying, we constructed and evaluated three kinds of random forest models, i.e., base model (BM), independent model (IM), and transfer model (TM) (Fig. S3; for details, refer to supplemental materials). We assessed these models (BM, IM, and TM) with 10-fold cross-validation. The IM had a mean area under the receiver operating characteristic curve (AUROC) of 0.768, which was higher than that of the BM (mean AUROC = 0.717) (Fig. 3A). However, in certain cities (such as Fairbanks, Barcelona, and Zurich), the performance of IM was inferior to that of BM (Fig. 3B). This phenomenon may originate from the sample size: cities with a small sample size do not have enough information to train a random forest model, which restricts the accuracy of the city classification model. The TM had a

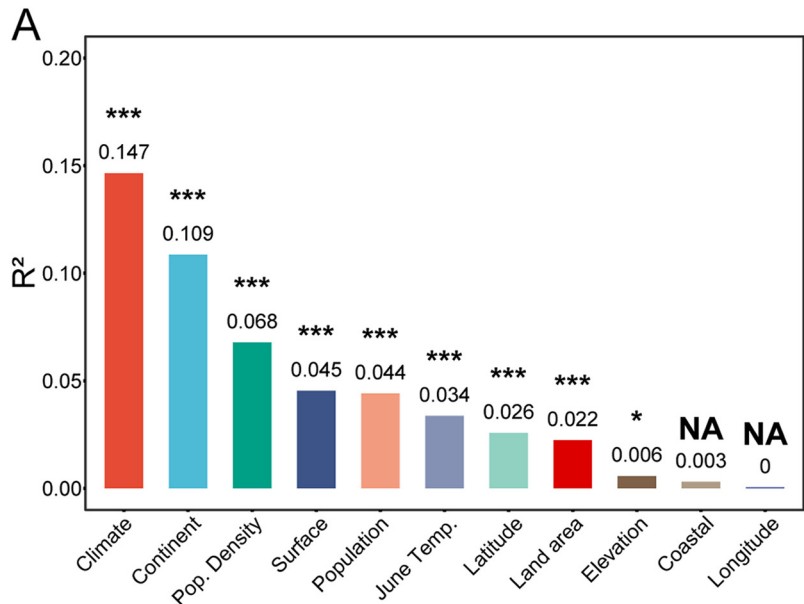

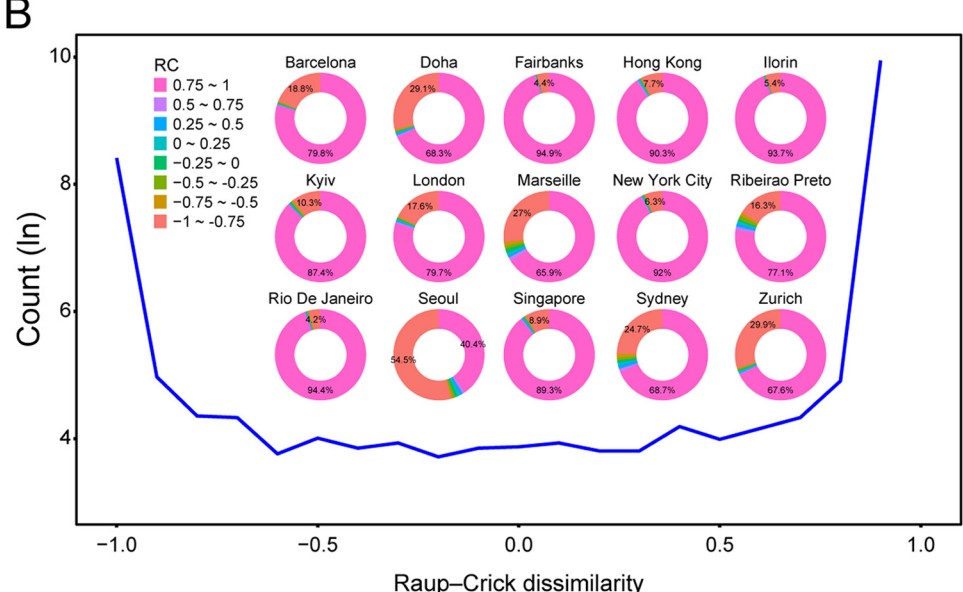

**FIG 2** Association between the stochastic ratio of urban transit system microbiome assemblage and city characteristics. (A) Bar plot showing the influencing factor ($x$ axis) that determines the characteristics of the urban transit system microbiome effects ($R^2$ of PERMANOVA, $y$ axis) for modified stochasticity ratio (MST) using PERMANOVA. Abbreviations: ***, $P < 0.001$; **, $0.001 \leq P < 0.01$; *, $0.01 \leq P < 0.05$; NS, nonsignificant difference. (B) Plot showing the $\beta_{RC}$ value ($x$ axis) distribution (the natural logarithm of $\beta_{RC}$ number, $y$ axis) of the urban transit system surface microbial community structure on a global scale. The inset pie charts show the $\beta_{RC}$ value distribution of the urban transit system surface microbial community structure in 15 cities. Color and number in the pie chart represent the grades of $\beta_{RC}$ value and the proportions of different grades, respectively.

mean AUROC of 0.782, which that was higher than that of the BM, and the performance of TM was similar to that of the IM in most cities or even better in some cities with small sample numbers, such as Fairbanks, Barcelona, and Zurich (Fig. 3B). This result indicates that adaptation of the BM to the newly introduced data reduced the influence of data heterogeneity from different surface types on the city classification analyses and provided more information for model learning. Thus, TL could further improve the accuracy of random forest classification models for quantifying the city specificity of UTS microbes.

Third, to quantify the city specificity of UTS microbes, we trained city classification models based on transfer random forest modeling using genera as classification features.

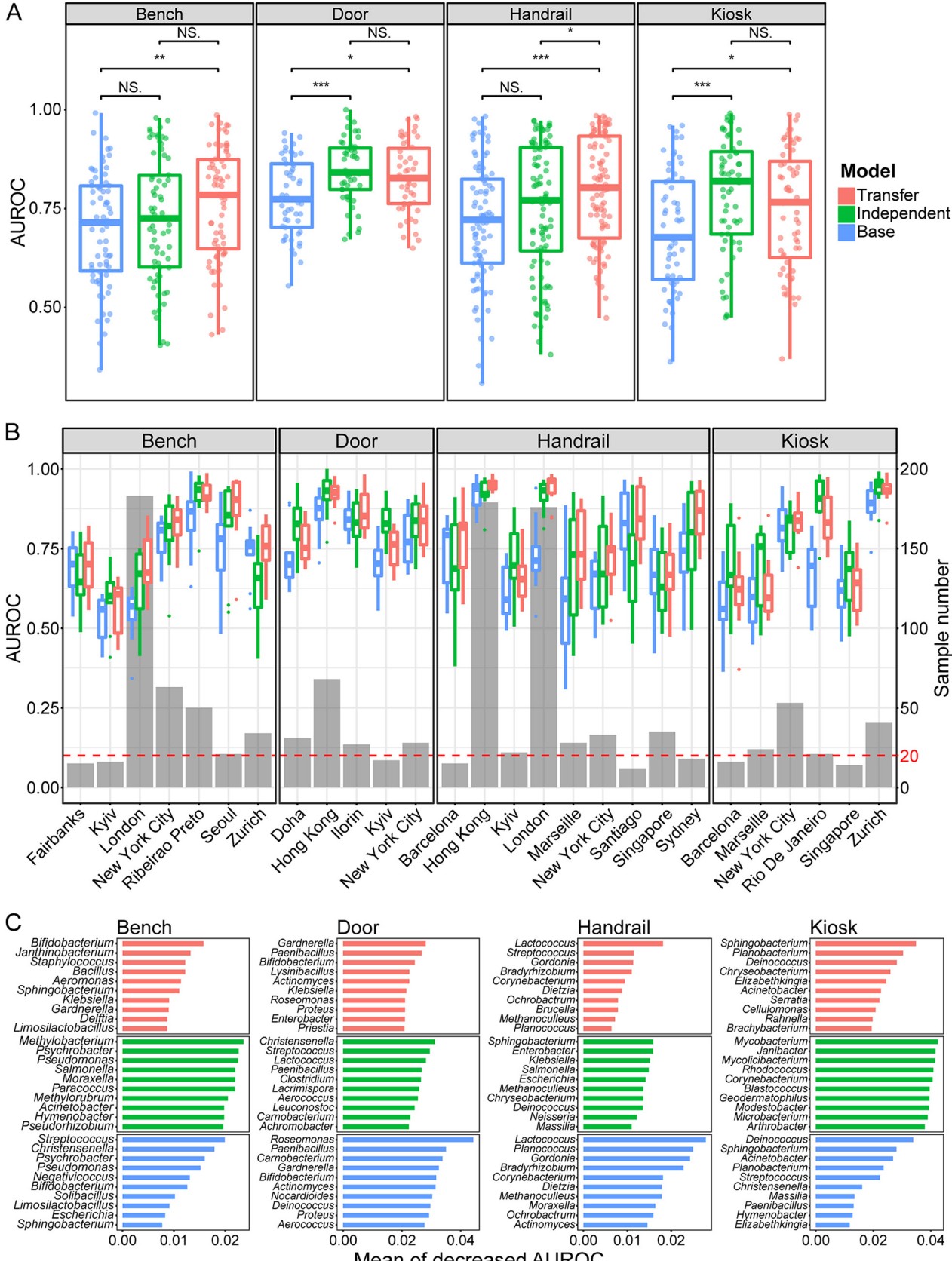

**FIG 3** Comparison of the performance of three kinds of random forest models and the mining of city-specific microbes of urban transit system surface microbial communities. (A) Box plot showing the performance (validation AUROC, *y* axis) of the base model, independent model, and

We defined the city specificity of the microbe as a mean decrease in the TM AUROC, i.e., the negative impact on model performance by excluding a feature. We selected 100 genera with the highest square deviation in relative abundance per surface data sets (bench, door, handrail, and kiosk) before quantifying the microbe city specificity. The microbes which have the top 10 city specificities per surface type vary across transit system surface types, and only certain genera were common on two surface types, including *Bifidobacterium*, *Gardnerella*, *Klebsiella*, and *Sphingobacterium*; however, no one was common among three or four transit system surface microbial communities (Fig. 3C). The microbes with high city specificity are dominated by soil and human microbes such as *Corynebacterium* (26), *Staphylococcus* (26), *Streptococcus* (26), and *Lactococcus*, and most of the microbes are bacteria except *Methanoculleus* (*Archaea*), which have been reported to dominate in both the stable and the deteriorative states of anaerobic digestion (37) and also were detected in the human oral microbiome (38). Thus, the *Methanoculleus* found on the handrail surface may shift from human saliva by human hand. We also found the UTS surface microbes with high city specificity have specific distribution patterns (Fig. S4), and that led us to assume that these microbes are correlated with city characteristics.

**Associations between UTS surface microbes and city characteristics.** We next investigated whether and to what extent city characteristics contributed to variations in UTS microbiota. To this end, we used the Mantel test to resolve the relationships between community diversity and city characteristics, as well as the correlation between the city-specific microbes and city characteristics. The UTS microbes which have the top 10 city specificities per surface type were designated city-specific microbes. We found that the contribution of city characteristics to the city-specific microbes varied across the four main surface types; all associations between the door and handrail surface microbial community and city characteristics were significant, compared with only two or three significant associations for other surfaces (Fig. 4). Significant associations between the transit system microbial community composition and city characteristics were also observed, which indicates that the city characteristics are associated with the transit system microbiome composition and city-specific microbes. We also found some pathogenic city-specific microbes (such as *Acinetobacter*, *Bacillus*, *Streptococcus*, and *Staphylococcus*) that correlated with city characteristics such as elevation, region, and average June temperature (Fig. S5). The correlation between pathogens and the city characteristics indicated the potential risk of infectious disease transmission in specific cities that may ensure that microbial-aware urban planning is more comprehensive.

To further confirm that the city specificity of UTS surface microbes strongly correlated with the city characteristics, we performed city characteristic classification analyses. We constructed a city characteristic (including elevation, latitude, longitude, population, proximity to the coast, population density, region, average June temperature, and Koppen climate) classification model using the microbes with the top 50 city specificities per surface type (bench, door, handrail, and kiosk) as features. While not all cities were equally classifiable (Fig. 5A) by using different microbes as features that were mined from different surface types (features for several cities, such as Fairbanks, Kyiv, and Singapore, which had a relatively low number of samples [<20] on different surface types, could not be classified effectively), in general, the predictions exceeded the classification using the other genera by a significant margin (Fig. S6).

The successful classification of city characteristics demonstrated distinct city-specific trends in the detected taxa and city metadata, as well as an improvement of the city

**FIG 3** Legend (Continued)
transfer model (*x* axis), using all genera as classification features and assigning samples to correct cities. Abbreviations: ***, $P < 0.001$; **, $0.001 \leq P < 0.01$; *, $0.01 \leq P < 0.05$; NS, nonsignificant difference. (B) Box plot showing the performance (validation AUROC, *y* axis) of the base model, independent model, and transfer models on four main surface types, using all genera as classification features and assigning samples to correct cities, in different cities (*x* axis). The bar plot shows the number (second *y* axis) of different surface samples in each city (*x* axis). (C) Bar plot showing the importance (quantified by mean decreased AUROC of the models with and without the genera, *x* axis) of the 10 most important features (*y* axis), mined by the base model, independent model, and transfer model, in four main urban transit system surface microbial community, respectively. One hundred genera with the highest variance in four surface samples were selected, respectively, before microbial mining. The data configuration, model training processes, and testing processes were repeated 10 times.

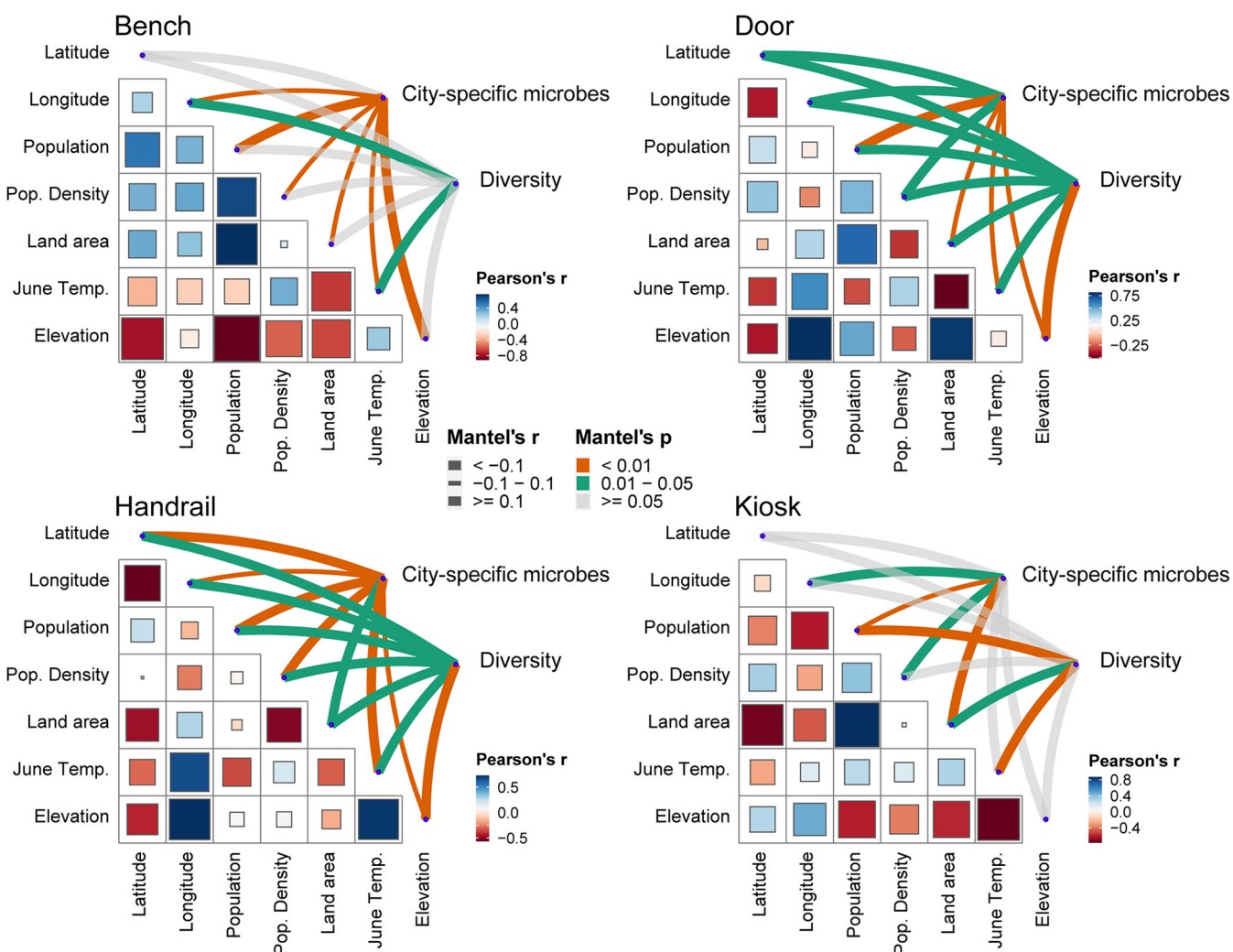

**FIG 4** Both city-specific microbes, as well as transit system surface microbial community compositions, were associated with city characteristics. The plot shows the correlations between the diversity (Shannon diversity index) of the urban transit system surface community and city characteristics and the correlations between the city-specific microbes' relative abundance and city characteristics. City characteristics include the latitude, longitude, population, population density, region, average temperature in June, and elevation above sea level. The correlations were calculated by Mantel test. The edge width corresponds to the *r* value of Mantel test, and the edge color denotes the statistical significance. Pairwise correlations of these variables are shown with a color gradient denoting Pearson's correlation coefficients. The orange line indicates a *P* value of less than 0.01, the green line indicates a *P* value between 0.01 and 0.05, and the gray line indicates a *P* value greater than 0.05.

characteristic classification model based on transfer learning (Fig. 5B). High accuracy of city identification may enable future forensic biogeographical capacities. For example, the surface of a person's bag might represent the "microbiota history" of that person's daily or weekly global travels, and the microbial data can potentially define the geo-spatial-genetic history of a person in many cities, as well as his or her pathogen risk or threat. These applications of public genetic data create potentially ambiguous ethical situations, whereby one's metagenome may hold clues about geospatial-genetic history, which reduces one's expectation of privacy, but they could also provide new forensic tools and methods for criminal justice as well as new mechanisms for disease and threat surveillance that are needed in increasingly urbanized human societies (11). The high accuracy of the city characteristic classification model further demonstrated the strong correlation between city-specific microbes and city characteristics.

## DISCUSSION

UTSs, especially urban subways and buses, are unique microbial environments with high residential density, diversity, and turnover, so they are of particular relevance to

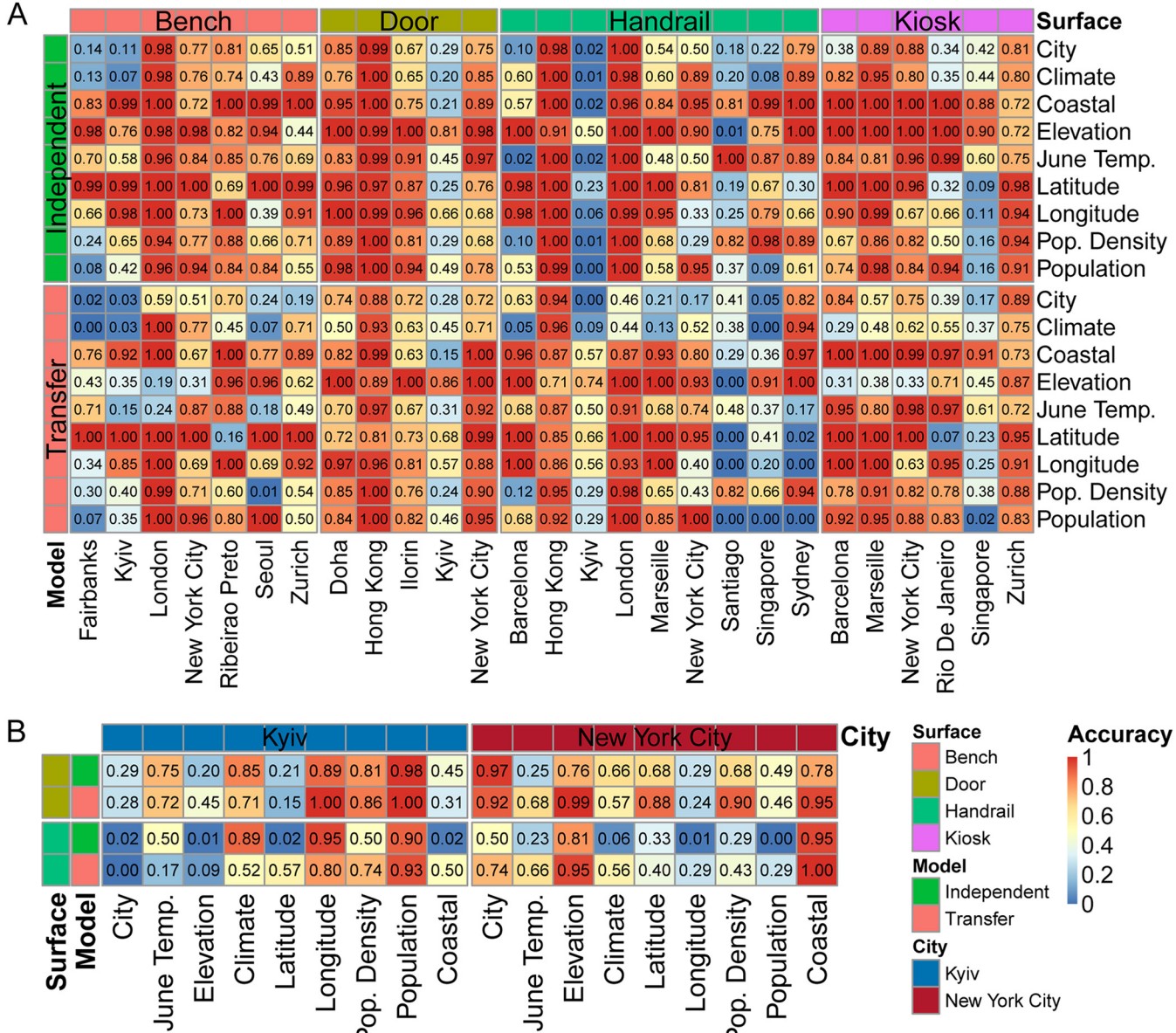

FIG 5 Random forest prediction models, using city-specific microbes as features, for city characteristics. (A) The heatmap shows the mean prediction accuracy of the independent model and transfer model, using the 50 most important city-specific microbes mined by the transfer model in each main surface as classification features, for a given feature (rows) in samples from a city (columns). Each main surface data set was randomly divided into the training set and testing set (20%/80%). The base models were trained using the other three main surface data. The data configuration, model training processes, and testing processes were repeated 10 times. The color of the row represents the type of prediction model; the color of the column represents the surface type which city-specific microbes that were used in the model training process were mined from. (B) The heatmap shows the mean prediction accuracy of the independent and transfer models (rows), using the 50 most important city-specific microbes mined by the transfer model in door and handrail surface (rows) as classification features, for given features (columns) in samples from Kyiv and New York City. The color of the row represents the surface type which city-specific microbes that were used in the model training process mined from; the color of the column represents the city of predicted samples. The data configuration, model training processes, and testing processes were repeated 10 times. Continuous features (e.g., population) were discretized to 3 bins by the pandas package in python {e.g., pandas.cut[city_characteristic, 3, labels=('low', 'medium', 'high')]}.

public health (12). Microbiota on urban transit surfaces are maintained by metapopulations of human skin symbionts and environmental generalists. Local interactions with the human body and environmental exposure drive the enrichment of microbial communities on transit surfaces. Thus, understanding the UTS microbiota assemblage is fundamental for understanding the relationship between the UTS microbiota and health features. While the neutral community model successfully estimated most of the relationships between the occurrence frequency of taxa and changes in their relative abundance in the UTSs, it is worth noting that city-specific microbe assemblies

were dominated by niche-based processes. We observed that ecological filters dominated the specific microbes of each UTS; these ecological filters were strongly associated with different city characteristics. These results show that the unique characteristics of cities shape the microbial communities and specific microbes of UTSs and that city-specific microbial assemblies are dominated by niche-based processes strongly associated with city characteristics. The microbes with high city specificity of UTS surfaces exhibited intercity heterogeneity. First, this may be explained by differences in the regional species pool within cities. Kim et al. (39) reported that a smaller species pool was observed in megacities, which suggests that city size and urbanization influence the microbial communities in UTSs. Second, the Mantel test showed that other city characteristics, including latitude, longitude, elevation, and land area, shaped the city-specific microbes. More importantly, city characteristics were accurately predicted by the abundance of the microbes with high city specificity that we mined.

An improved understanding, based on ecological theory, of how UTS microbiomes are assembled could facilitate public health monitoring. If pathogenic infection in humans occurs and spreads through a UTS and the infection is hindered or regulated by specific microbial community assemblages, this approach may also provide a solution for treating infectious diseases. Our investigation of the UTS microbiome and its assemblages in this study can serve as a cornerstone for human urban transportation system service management. The specific microbes of a UTS potentially associate the possible outbreak of infection or disease in a city, and an in-depth understanding of the main processes of the assemblage of city-specific microbes can provide guidance for clinicians to treat diseases and provide personalized treatment in cities with characteristic differences.

Cities generally have an impact on human health, although the mechanisms of this impact are widely variable and often poorly understood. Our study showed that microbes on the surface of UTSs were pathogenic microbes. According to a study published in 2002, an estimated 10% to 13% of farm animals are infected with *Brucella* species (40). Certain species of *Brucella*, e.g., *B. melitensis* and *B. suis*, can cause nonspecific symptoms (flulike manifestations) that are especially dangerous to humans (41). We found *Brucella* abundant in east Asia (see Fig. S4 in the supplemental material) and remind people to pay attention to the transmission of specific bacteria in a transit system. If regular sampling of urban transportation system surfaces is performed, it is possible to identify and quantify circulating infectious microorganisms, helping to identify and predict disease outbreaks. Consequences of the deterministic assemblage may not be represented only on subway railings or handrails but also on skin that directly touches surfaces in UTSs, and even other human organs that host microbiota (e.g., the gastrointestinal tract) (39). However, urban characteristics affect the formation of microbiota on surfaces of UTSs and even human organs. Urban transport systems harbor rich biodiversity and novel molecules that can be used for treatment, and we suggest that large-scale disease outbreak monitoring and prediction may require specific approaches based on differences in the microbiome of urban transport systems.

**Conclusions.** This study explored the assemblage of UTS microbiota, as well as the strong correlation between city characteristics and UTS surface microbiota. First, the stochastic process plays a dominant role in the assemblage of UTS microbiota and the city characteristics shaping the stochastic ratio. Second, the city specificity of UTS microbes was quantified by using a machine learning model based on transfer learning and random forest. Third, the strong association between city characteristics and the UTS microbiota was identified, based on which city characteristics can be accurately predicted by using city-specific microbes. The representative example was New York City, for which the NCM explanation rate of 0.831 indicated stochastic processes dominant in UTS microbiota assemblage. However, the accuracy of the city characteristic classification model was over 0.9, suggesting a strong association between city characteristics and UTS microbes. These patterns can facilitate a broad spectrum of applications, including public health monitoring, as well as city-specific and microbial-aware urban planning.

## MATERIALS AND METHODS

**Data set.** A total of 3,359 surface samples (assigned taxonomic table and metadata were downloaded at https://pngb.io/metasub-2021) were collected and analyzed by the MetaSUB project (24) across 16 cities worldwide (see Fig. S1 in the supplemental material) at three major time points, a pilot study in 2015 to 2016 and 2 global city sampling days (21 June) in 2016 and 2017. These samples mainly originated from four surface types in different urban transit systems (UTSs), including bench, door, handrail, and kiosk ($n = 1,240$) (Table S1). Here, we present the results and code of our analyses to access, a GitHub repository (https://github.com/XiongGZ/UTS_Microbiota).

**Analysis of the microbial community assemblage.** To determine the importance of stochastic processes in the assemblage of transit system microbial communities, the neutral community model (NCM) (42) was used to explore the relationship between the occurrence frequency of genera and their relative abundance with the influence of dispersal. The R code was downloaded from GitHub (https://github.com/Weidong-Chen-Microbial-Ecology/Stochastic-assembly-of-river-microeukaryotes) (43).

The modified stochasticity ratio (MST) was used to determine the roles of the deterministic and stochastic processes in the assemblage of a microbial community, which were calculated using the NST package in R (44). The value of the MST reflects the importance of stochastic processes in the community assemblage: an absolute value over 0.5 indicates more stochastic processes than deterministic processes, and less than 0.5 indicates that more deterministic processes contribute to the microbial community assemblage (33).

To determine the degree to which a community created by deterministic processes deviates from a community based on random zero expectation, the modified Raup-Crick dissimilarity metric was calculated using the NST package in R (https://github.com/DaliangNing/NST) (44). The Raup-Crick index indicates the potential mechanism of community construction: a value of 0 indicates that the observed degree of similarity (or dissimilarity) is not different from zero expectation, 1 indicates that the observed difference is higher than the zero expectation of any simulation (the difference between communities is completely greater than the accidental expectation), and $-1$ indicates that the observed difference is lower than the zero expectation of any simulation (the difference between communities is completely less than the accidental expectation).

**Artificial intelligence based on random forest approaches.** Random forest (RF) approaches are robust ensemble machine-learning methods for classification and regression. However, region limitations, such as small sample number, exist in these methods when quantifying the ability of UTS microbes in discriminating between cities. To expand the universality of RF for quantifying, the model-based transfer learning approach was used to adapt knowledge for the new data set. The TransferRandomForest packages in python (https://github.com/Luke3D/TransferRandomForest) (45) were used to construct classification models of correct cities for quantifying. The detailed data configuration, model construction, and validation are described in the supporting information. The ability of UTS microbes in discriminating between cities was quantified using the mean decrease in city classification model AUROC, i.e., the negative impact on model AUROC by excluding a feature.

**Analysis of microbial source tracking.** To determine the association of UTS microbial communities, the FEAST (46) model using default parameter settings was applied for calculating the contribution of different surface microbial communities in the New York City transit system. The New York City transit system has the largest mass transit system in the world and had 1.1 billion unlinked transit passenger trips in 2021 (28) and more samples from different surfaces than other cities. Compared to the commonly used Markov chain Monte Carlo (MCMC)-based Bayesian SourceTracker approach (47), FEAST is 30 to 300 times faster and more accurate when the target microbial community contains taxa from an unknown, uncharacterized source. These advantages facilitate the usage of FEAST in fractioning the proportion of each source in the sink. A bidirectional analysis was performed to investigate the associations between different surface types of microbes by assuming either the microbiota in one surface as a sink and the others as sources.

**Analysis of the effect of city characteristics on microbial profiles.** To determine the relationships between the city characteristics (including the continent, latitude, longitude, population, population density, surface type, elevation, proximity to the coast, region, average June temperature, and Koppen climate, which were also recorded for each city) and the microbes on UTS surfaces, the Mantel test, based on using the Spearman method and performed using the linkET package in R (48), was used to calculate the association between UTS surface microbes and city characteristics.

**Statistical analyses.** Reads not assigned to the phylum level were removed before analysis. Alpha diversity (Shannon index) was calculated using the vegan package in R (49). Principal-coordinate analysis (PCoA) was used to visualize the relationships among the communities of different samples using the Bray-Curtis index (vegan and ape packages in R [49, 50]). Permutational multivariate analysis of variance (PERMANOVA) analyses were performed to evaluate the difference between different surface types of microbial communities using the Bray-Curtis index (vegan package in R [49]).

## SUPPLEMENTAL MATERIAL

Supplemental material is available online only.

**SUPPLEMENTAL FILE 1**, PDF file, 4.3 MB.

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
