## [Reviewer comments · Microbiology Spectrum]

Microbiology Spectrum

Niche-based microbial community assemblage in urban transit systems and the influence of city characteristics

Guangzhou Xiong, Lei Ji, Mingyue Cheng, and Kang Ning

Corresponding Author(s): Kang Ning, Huazhong University of Science and Technology

Review Timeline:

Submission Date:	January 10, 2023
Editorial Decision:	January 29, 2023
Revision Received:	February 7, 2023
Accepted:	February 10, 2023

Editor: Feng Gao

Reviewer(s): Disclosure of reviewer identity is with reference to reviewer comments included in decision letter(s). The following individuals involved in review of your submission have agreed to reveal their identity: Jialiang Yang (Reviewer #1)

Transaction Report:

DOI: <https://doi.org/10.1128/spectrum.00167-23>

January 29, 2023

Prof. Kang Ning
Huazhong University of Science and Technology
School of Life Science and Technology
1037 Luoyu Road, HUST East Campus. Building 11, Room 504
Room 504, East 11 Building
Wuhan, Hubei 430074
China

Re: Spectrum00167-23 (Niche-based microbial community assemblage in urban transit systems and the influence of city characteristics)

Dear Prof. Kang Ning:

Link Not Available

Sincerely,

Feng Gao

Journals Department
Reviewer comments:

Reviewer #1 (Comments for the Author):

Microbial community profile and assemblage in urban transit systems and the influence of city characteristics is an important yet challenging topic. The study has novelty in that it used ecological theories and transfer learning. If community patterns were predictable at the local or the city scale, the manuscript suggests this understanding could facilitate a broad spectrum of applications including public health monitoring or city-specific microbial-aware urban planning. Overall, I find that the manuscript describes a thorough examination of the microbial communities in urban transit systems. The

idea that these systems can have characteristic or signature microbial communities is important, especially in terms of possible pathogen monitoring. I agree with the setup in the manuscript that a baseline understanding of the microbes that inhabit or persist in these spaces and the local factors that influence these assemblages is needed if a monitoring framework is to be implemented in the future. The results are comprehensive and reliable, the whole manuscript is well organized. In principle, this paper represents a timely work for the urban transit system microbiome.

Minor Issues

1. Was the FEAST model run with all default settings or were any settings changed? There is not enough detail to recreate the FEAST model runs. IF it is too detailed, then this could go to supplemental methods.
2. Figure 2 legend title is a more a result statement then a short description of what is being presented. This should be changed to be more descriptive of what data the figure displays.

Reviewer #2 (Comments for the Author):

Xiong et al reported the microbial community assemblage in urban transit systems and the factors effected it by AI. The work is well-done, and I have only few questions on writing. The author should completely check the MS, especacially the usage of abbreviation.

Line 31, which characteristics can affect should be explained.

Line 68 , COVID-19 should be written in the full name.

Line78, if there are multiple factors, it should not only be two.

Line 156, line 75, and line 204, the explanations in the two parts are repeated.

Line 195, 186, the abbreviation of UTS has been mentioned already.

Staff Comments:

Preparing Revision Guidelines

Please return the manuscript within 60 days; if you cannot complete the modification within this time period, please contact me. If you do not wish to modify the manuscript and prefer to submit it to another journal, please notify me of your decision immediately so that the manuscript may be formally withdrawn from consideration by Microbiology Spectrum.

Comments

Microbial community profile and assemblage in urban transit systems and the influence of city characteristics is an important yet challenging topic. The study has novelty in that it used ecological theories and transfer learning. If community patterns were predictable at the local or the city scale, the manuscript suggests this understanding could facilitate a broad spectrum of applications including public health monitoring or city-specific microbial-aware urban planning.

Overall, I find that the manuscript describes a thorough examination of the microbial communities in urban transit systems. The idea that these systems can have characteristic or signature microbial communities is important, especially in terms of possible pathogen monitoring. I agree with the setup in the manuscript that a baseline understanding of the microbes that inhabit or persist in these spaces and the local factors that influence these assemblages is needed if a monitoring framework is to be implemented in the future. The results are comprehensive and reliable, the whole manuscript is well organized. In principle, this paper represents a timely work for the urban transit system microbiome

Minor Issues

1. Was the FEAST model run with all default settings or were any settings changed? There is not enough detail to recreate the FEAST model runs. IF it is too detailed, then this could go to supplemental methods.

2. Figure 2 legend title is a more a result statement then a short description of what is being presented. This should be changed to be more descriptive of what data the figure displays.

Response to Comments from Reviewers

Dear Reviewers,

We greatly appreciate the efforts of you for reviewing and providing constructive suggestions on our manuscript (Manuscript Number: Spectrum00167-23). We have studied the comments carefully and have made revision according to all comments. All of the revisions are highlighted in red in the revised manuscript. The point-by-point answers to the comments and suggestions were listed as below.

Reviewer #1:

Microbial community profile and assemblage in urban transit systems and the influence of city characteristics is an important yet challenging topic. The study has novelty in that it used ecological theories and transfer learning. If community patterns were predictable at the local or the city scale, the manuscript suggests this understanding could facilitate a broad spectrum of applications including public health monitoring or city-specific microbial-aware urban planning. Overall, I find that the manuscript describes a thorough examination of the microbial communities in urban transit systems. The idea that these systems can have characteristic or signature microbial communities is important, especially in terms of possible pathogen monitoring. I agree with the setup in the manuscript that a baseline understanding of the microbes that inhabit or persist in these spaces and the local factors that influence these assemblages is needed if a monitoring framework is to be implemented in the future. The results are comprehensive and reliable, the whole manuscript is well organized. In principle, this paper represents a timely work for the urban transit system microbiome.

Answer: We thank reviewer for these insights. We have made the explanations about the urban transit system microbiome in more details and with higher clarity, as shown in main text and below answers.

Minor Issues

1. Was the FEAST model run with all default settings or were any settings changed? There is not enough detail to recreate the FEAST model runs. IF it is too detailed, then this could go to supplemental methods.

Answer: We thank reviewer for this comment. The FEAST model run with all default settings. The source and sink used in source tacking analysis were described in Results part (Line 206-208: ‘Microbial communities from one surface type were used as the sinks, and the samples from 13 other surface types were used as the sources.’). To make this clearer, we have updated the Method part in main text as below (Line 153-155):

“To determine the association of UTS microbial communities, the FEAST (Shenhav et al., 2019) model using default parameter settings was applied for calculating the contribution of different surface microbial communities in New York City transit system.”

2. Figure 2 legend title is a more a result statement then a short description of what is being presented. This should be changed to be more descriptive of what data the figure displays.

Answer: We thank reviewer for this comment. We have updated Figure 2 legend title in Figure captions part as below (Line 735-737):

“The association between the stochastic ratio of urban transit system microbiome assemblage and city characteristics.”

Reviewer #2:

Xiong et al reported the microbial community assemblage in urban transit systems and the factors effected it by AI. The work is well-done, and I have only few questions on writing. The author should completely check the MS, especially the usage of abbreviation.

Answer: We thank reviewer for these insights. We have made the explanations about the urban transit system microbiome in more details and with higher clarity, as shown in main text and below answers.

Line 31, which characteristics can affect should be explained.

Answer: We thank reviewer for this comment. We have updated the Abstract part in main text as below (Line 29-31):

“Furthermore, by utilizing an artificial intelligence model, we quantified the ability of UTS microbes in discriminating between cities, and found that the ability was also strongly affected by city characteristics, especially climate and continent.”

Line 68, COVID-19 should be written in the full name.

Answer: We thank reviewer for this comment. We have updated the Introduction part in main text as below (Line 68):

“For example, Corona Virus Disease 2019 (COVID-19) asymptomatic subjects could mass transmit the pathogenic SARS-CoV-2 (severe acute respiratory syndrome coronavirus) in UTS by breathing (Turnbaugh et al., 2007).”

Line78, if there are multiple factors, it should not only be two.

Answer: We thank reviewer for this comment. We have updated the Introduction part in main text as below (Line 77-79):

“The diversity, composition, and stability of UTS microbiota are influenced by multiple factors (Lax et al., 2014), including surface type (Hsu et al., 2016) and city characteristics (including latitude, longitude, altitude and climate).”

Line 156, line 75, and line 204, the explanations in the two parts are repeated.

Answer: We thank reviewer for this comment. We have deleted the explanations the Introduction part (Line 75-76) and Results part (Line 202-204) in main text as below.

Introduction part (Line 75-76)

“The United States had 4.7 billion unlinked transit passenger trips in 2021 (January 20–December 20), and New York City, which has the largest mass transit system in the world, had 1.1 billion unlinked transit passenger trips in 2021 (APTA Ridership Report - Fourth Quarter 2021 Ridership,

n.d.).”

Results part (Line 202-204)

~~“New York City has the largest mass transit system in the world which had 1.1 billion unlinked transit passengers trips in 2021 (APTA Ridership Report—Fourth Quarter 2021 Ridership, n.d.). The samples used in source tracking analyses were collected from New York City UTS surface types (N = 402, including 14 different surface types, Table S2).”~~

Line 195, 186, the abbreviation of UTS has been mentioned already.

Answer: We thank reviewer for this comment. We have used the abbreviation of UTS in the Results part in main text as below.

Line 185: “3.1 Assemblage of UTS microbiota”

Line 249-250: “3.2 Quantify the city specificity of UTS microbes using artificial intelligence”

Line 303-304: “3.3 Associations between UTS surface microbes and city characteristics”

February 10, 2023

Prof. Kang Ning
Huazhong University of Science and Technology
School of Life Science and Technology
1037 Luoyu Road, HUST East Campus. Building 11, Room 504
Room 504, East 11 Building
Wuhan, Hubei 430074
China

Re: Spectrum00167-23R1 (Niche-based microbial community assemblage in urban transit systems and the influence of city characteristics)

Dear Prof. Kang Ning:

Your manuscript has been accepted, and I am forwarding it to the ASM Journals Department for publication. You will be notified when your proofs are ready to be viewed.

Sincerely,

Feng Gao
Editor, Microbiology Spectrum
